# Early Pandemic Evaluation and Enhanced Surveillance of COVID-19 (EAVE II): protocol for an observational study using linked Scottish national data

Colin R Simpson,[1,2] Chris Robertson,[3,4] Eleftheria Vasileiou [ID],[2] Jim McMenamin,[4] Rory Gunson,[5] Lewis D Ritchie,[6] Mark Woolhouse,[2] Lynn Morrice,[2] Dave Kelly,[7] Helen R Stagg [ID],[2] Diogo Marques [ID],[4] Josie Murray,[4] Aziz Sheikh[2]

For numbered affiliations see end of article.

**Correspondence to**
Dr Eleftheria Vasileiou;
eleftheria.vasileiou@ed.ac.uk

## ABSTRACT

**Introduction** Following the emergence of the novel severe acute respiratory syndrome coronavirus 2 (SARS-CoV-2) in December 2019 and the ensuing COVID-19 pandemic, population-level surveillance and rapid assessment of the effectiveness of existing or new therapeutic or preventive interventions are required to ensure that interventions are targeted to those at highest risk of serious illness or death from COVID-19. We aim to repurpose and expand an existing pandemic reporting platform to determine the attack rate of SARS-CoV-2, the uptake and effectiveness of any new pandemic vaccine (once available) and any protective effect conferred by existing or new antimicrobial drugs and other therapies.

**Methods and analysis** A prospective observational cohort will be used to monitor daily/weekly the progress of the COVID-19 epidemic and to evaluate the effectiveness of therapeutic interventions in approximately 5.4 million individuals registered in general practices across Scotland. A national linked dataset of patient-level primary care data, out-of-hours, hospitalisation, mortality and laboratory data will be assembled. The primary outcomes will measure association between: (A) laboratory confirmed SARS-CoV-2 infection, morbidity and mortality, and demographic, socioeconomic and clinical population characteristics; and (B) healthcare burden of COVID-19 and demographic, socioeconomic and clinical population characteristics. The secondary outcomes will estimate: (A) the uptake (for vaccines only); (B) effectiveness; and (C) safety of new or existing therapies, vaccines and antimicrobials against SARS-CoV-2 infection. The association between population characteristics and primary outcomes will be assessed via multivariate logistic regression models. The effectiveness of therapies, vaccines and antimicrobials will be assessed from time-dependent Cox models or Poisson regression models. Self-controlled study designs will be explored to estimate the risk of therapeutic and prophylactic-related adverse events.

**Ethics and dissemination** We obtained approval from the National Research Ethics Service Committee, Southeast Scotland 02. The study findings will be presented at international conferences and published in peer-reviewed journals.

## Strengths and limitations of this study

► We plan to interrogate national data on the Scottish general population.
► We are expanding an existing national pandemic reporting platform, which uses anonymised individual patient-level data from general practices, hospitals, death registry, virology (reverse transcriptase PCR) and serology tests to investigate the epidemiology of COVID-19 and assess the effectiveness of existing or future preventive and treatment measures.
► This is an observational study; therefore, insufficient adjustment for confounding, either due to insufficiently granular variable measurement or a lack of variable measurement, is a potential concern.

## INTRODUCTION

In the last two centuries, six pandemics (global epidemics) have emerged due to novel influenza and coronavirus strains. During the 20th century, influenza caused three pandemics (1918–1919, 1957–1958 and 1968–1969), resulting in millions of clinical cases and deaths.[1–4] An estimated 20–50 million deaths were reported during the 1918–1919 influenza pandemic. Fewer (between 1 million and 4 million deaths) were estimated for the 1957–1958 and 1968–1969 influenza pandemics, respectively.[1–4] The high mortality rates observed in the 20th century against the H1N1, H2N2 and H3N2 influenza viruses were mainly due to lack of prophylactic and therapeutic interventions, such as influenza vaccines and antiviral medications.[1–4] By comparison, the first pandemic of the 21st century arose from a novel coronavirus, severe acute respiratory syndrome (SARS-CoV), which emerged in 2002–2003.[5] SARS caused more than 8000 infections and

700 deaths globally.[2 5] In 2009–2010, the fourth recorded influenza pandemic, influenza A (H1N1), emerged in Mexico, resulting in more than 200 000 deaths globally. Approximately 11%–21% of the global population was infected.[2 6] Previous exposure to seasonal influenza vaccination induced little or no cross-reactive antibody responses.[7] Particularly low immunological protection against the virus was observed in the younger population (<30 years old) compared with older adults.[7]

In December 2019, a novel coronavirus—SARS coronavirus 2 (SARS-CoV-2)—emerged in Wuhan, China.[8 9] In the space of 4 months, this virus has now spread globally. The World Health Organisation (WHO) declared the coronavirus outbreak a Public Health Emergency of International Concern on 30 January 2020 and then a pandemic on 11 March 2020, as a result of the worldwide spread of the COVID-19 disease.[9] As of 3 April 2020, the WHO has reported more than 970 000 confirmed infections globally and over 50 000 deaths.[9] The elderly, people with underlying medical conditions and people with poor immune function and long-term users of immunosuppressive agents are particularly vulnerable to SARS-CoV-2 and at risk of severe coronavirus-related illness.[8–11] Current data suggest that SARS-CoV-2 has a lower mortality rate, ranged between 0.25% and 3%, than for SARS-CoV (10%) and Middle East Respiratory Syndrome-related coronavirus (MERS-CoV) (37%), respectively.[12 13] It has been postulated (using data from case studies) that the main driver of disease severity among younger patients for COVID-19 are immunopathological lesions, resulting from an excessive proinflammatory host response or cytokine storm.[14 15] Among older people, an impaired interferon pathway and systemic virus dissemination beyond the respiratory tract may lead to severe disease.[14 15] The absence of immunity from historic exposure to existing seasonal vaccination or antiviral therapy also (in comparison with influenza) renders COVID-19 a significant global health threat, which demands an urgent response from national and international agencies.

Rapid, large observational epidemiological studies are now required to identify the epidemiological and clinical profile of the COVID-19 pandemic. These studies can also be used to estimate the effectiveness of any existing or new healthcare interventions, such as vaccines and antiviral therapies (eg, the introduction of any new pandemic vaccine), where it is unethical and/or not feasible to mount more rigorous experimental studies.

Using linked routine sources of primary, secondary, mortality and virological/serological testing data, this study aims to describe the epidemiology of COVID-19 in Scotland and in due course help establish the effectiveness of existing or new therapeutic interventions against the coronavirus that are not subjected to formal clinical trials. Specifically, our objectives are to:

### Primary objectives
a. Determine the epidemiological risk factors for infection, morbidity and mortality of COVID-19 (eg, laboratory and serology confirmed SARS-CoV-2 infection in relation to demographic, socioeconomic and clinical population characteristics).
b. Determine the healthcare burden of COVID-19 (eg, COVID-19 related morbidity and mortality in relation to demographic, socioeconomic and clinical population characteristics).

### Secondary objectives
a. Measure the uptake of prophylactic interventions (eg, vaccines).
b. Estimate the effectiveness of any new or existing prophylactic and therapeutic interventions (eg, new or repurposed therapies, vaccines and antimicrobials).
c. Assess the safety of any new or existing of prophylactic and therapeutic interventions (eg, new or repurposed therapies, vaccines and antimicrobials).

This work will repurpose and expand the hibernated Early Estimation of Vaccine and Anti-Viral Effectiveness (EAVE) project as part of the National Institute for Health Research (NIHR) Pandemic Preparedness Research Portfolio[16 17] and a proven platform for studies on seasonal and pandemic influenza vaccine and antiviral assessment.[17–21]

## METHODS
### Study design and population
We will undertake a timely analysis of a large national open prospective observational cohort of patients using a unique community, hospital and laboratory linked dataset. We will seek to extract data on 5.4 million people from across Scotland (figure 1). Therefore, our study

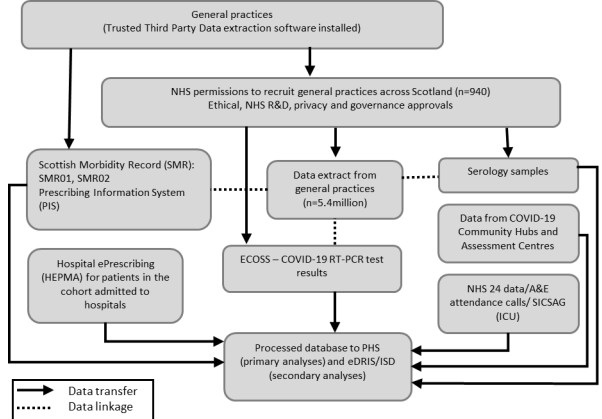

**Figure 1** Flow diagram for EAVE II project. A&E, Accident & Emergency; EAVE II, Early pandemic evaluation and enhanced surveillance of COVID-19; ECOSS, Electronic Communication of Surveillance in Scotland; eDRIS, the electronic data research and innovation service; HEPMA, Hospital Electronic Prescribing and Medicines Administration; ICU, intensive care unit; ISD, Information Services Scotland; NHS, National Health Service; PHS, Public Health Scotland; RT-PCR, reverse transcriptase PCR; R&D, Research & Development; SICSAG, Scottish Intensive Care Society Audit Group.

aims to collect data from all residents in Scotland registered with a general practice, which translates to over 91% coverage of the Scottish population.[21]

## Databases

Individual-level data from general practices will be extracted and linked deterministically to secondary and laboratory healthcare datasets using the Community Health Index (CHI).[17] The CHI number is a unique identifier provided by the National Health Service (NHS) for each resident in Scotland registered with a general practice. A CHI number is also allocated to patients that may have no number when present for treatment as the CHI number is mandatory for all clinical communications. Thus, non-Scotland resident patients and other temporary residents can also have a CHI number allocated, if required; however, wherever possible, temporary patients will be excluded from this analysis.[22] The linkage of the datasets and analysis will take place within a secure Trusted Research Environment.[17]

### Primary care

Almost all individuals in Scotland are registered with a general practice, which provide free of charge healthcare services. Data from all patients registered in general practices will be extracted and studied. The University of Edinburgh and Public Health Scotland (PHS) will recruit the additional general practices through Albasoft Ltd.[17–21] Albasoft Ltd is the trusted third party that will carry out the data extraction from all general practices using the Enhanced Services Contract Reporting Options system.[17–21] We will also extract data from a network of COVID-19 Community Hubs and Assessment Centres established by NHS Health Boards across Scotland.[23] The aim of this network is to provide a direct and rapid route of people with COVID-19 symptoms that have worsened or not improved after a week. Patients can call NHS 24 for an initial assessment and then if needed the call will be passed to a telephone community hub, staffed by clinical decision makers.[23] The clinical decision maker will then decide if an appointment for a face-to-face consultation at an assessment centre is necessary.[23] Previous observational studies have shown over 91% completeness of capture of contacts and accuracy of clinical event coding (Read codes) among practices in Scotland.[21]

### Secondary care

The Scottish Morbidity Record (SMR) database will be used to derive information for all in-patient hospitalisations and emergency admissions in Scotland, which is maintained by the Information Services Division.[24] Specifically, we will use data from the SMR01 record, which is an episode-based patient record for all inpatients and day cases discharged from non-obstetric and non-psychiatric specialties in Scotland.[25] Data from the SMR02 record will also be used, which is an episode-based patient record for all inpatients and day cases from obstetric specialties in the NHS Scotland.[26] The SMR dataset also contains

mortality data that derive from the National Records of Scotland.[27] Regular validation checks are applied to the SMR database. The latest data quality assessment of these SMR datasets have shown over 90% completeness and accuracy in consistency with previous years.[28] We will also extract and link data on prescribing and administration of medicines for inpatients that are available from Scottish Hospital Electronic Prescribing and Medicines Administration systems.[29] The study data will also be linked with data from patients admitted to adult general intensive care units (ICU), which derive from the Scottish Intensive Care Society Audit Group (SICSAG) national database.[30] The database contains detailed information on the management of critically ill or injured patients. Data are collected from all general ICU and combined ICU/high-dependency units (HDUs). Data from more than 90% of general HDUs and a number of specialist ICUs and HDUs are collected by the database.[30]

### Laboratory and serology data

The Electronic Communication of Surveillance in Scotland (ECOSS) system of PHS is a database that holds surveillance data on various microorganisms (eg, influenza virus and coronavirus) and infections reported from diagnostics and reference laboratories.[31] Data on laboratory results for all reverse transcriptase PCR (RT-PCR) tests carried out in Scotland are being collated by ECOSS and can be linked to other data sources.[31] Positive laboratory swab samples for SARS-CoV-2 will also be sent to national sequencing centres where 500 SARS-CoV-2 genome sequences will be performed.

In a substudy, the West of Scotland Specialist Virology Laboratory will collect and store residual sera from routine blood tests from patients until the serology test becomes available.[32] The EAVE study has already stored 1000 biochemistry samples from a subset of participating practices from 2014, demonstrating that a potential mechanism for the collection and storage of the residual sera works.[17] We aim to collect and store serially throughout the duration of the COVID-19 pandemic. This will be used to determine exposure to SARS-CoV-2 and other viruses by the presence of antibodies.[17]

### Exposure definitions and potential confounding factors

The following exposure variables will be used in relation to the study's primary outcomes: sex, age, socioeconomic status (SES) and clinical at-risk group. SES will be determined based on the Scottish Multiple Deprivation Index (SIMD). The SIMD classification is based on deprivation quintiles. Quintile 1 refers to the most deprived and quintile 5 refers to the least deprived. The SIMD is a combination of 38 indicators of the following seven domains: income, employment, health, education, housing, geographical access to services and crime.[21] Clinical at-risk groups refer to individuals with certain underlying medical conditions where are at-risk of COVID-19 related complications and for whom seasonal influenza vaccination is recommended. The following clinical at-risk

conditions will be considered: (A) chronic respiratory disease (with chronic obstructive pulmonary disease and asthma as subsets); (B) chronic heart disease; (C) chronic liver disease; (D) chronic kidney disease; (E) chronic liver disease; (F) chronic neurological disease; (G) diabetes types 1 and 2; (H) conditions or medications causing impaired immune function; (I) pregnancy; (J) asplenia or dysfunction of spleen; (K) obesity (body mass index (BMI) <20, 20–25, 25–30, 30–39 and $\geq 40\,kg/m^2$); (L) hypertension (subsets controlled/uncontrolled hypertension); (M) tuberculosis and (N) multimorbidity.[21] This list will be updated as more evidence arises within the medical literature. The following exposure variables will be used in relation to study's secondary outcomes: any new vaccines against SARS-CoV-2 and existing or new therapies and antimicrobial medication against COVID-19. These will be determined once our study data are available and any new therapies, vaccines and antimicrobials specifically against the SARS-CoV-2 virus have been produced.

A number of aforementioned and additional population characteristics below will also be used as potential confounding factors in relation to the study's primary and secondary outcomes. Charlson Comorbidity Index will represent the weighted comorbidity score based on secondary care data.[17–21] The urban/rural location will be determined based on the urban/rural eight-fold classification (UR8). The UR8 is the definition of rural areas in Scotland: 1 is assigned to large urban areas and 8 is assigned to remote rural areas.[21] Smoking status will be determined and presented into the following four categories: current smoker, non-smoker, ex-smoker and not recorded for patients with no data on smoking.[17–21] The type of smoking products (eg, vaping products) and alcohol use will also be determined, if possible. Previous healthcare usage will be used to measure number of primary care consultations and secondary care admissions in previous years. The number of prescriptions will also be determined for previous years.[17–21] General practice will also be used to account the effect of clustering within practices. The effect of population density will also be investigated. Additional exposures such as number of household members for those with a confirmed SARS-CoV-02 infection and daily protective measures will also be investigated given the high transmission rate of COVID-19.

## Outcome definitions

The primary outcomes of this study will include: (A) laboratory confirmed SARS-CoV-2; (B) serum from blood samples taken from biochemistry tests (or rapid antibody tests if available) will be used to determine exposure to SARS-CoV-2 infection by the presence of antibodies; and (C) SARS-CoV-2 infection related clinical outcomes including general practice, COVID-19 centres and out-of-hours consultations, hospital admissions including secondary bacterial infections and multidrug-resistant bacteria associated with these infections, emergency admissions, out of hours consultations and deaths. Secondary outcomes include: (A) vaccine uptake proportions; (B) prevention and reduction of SARS-CoV-2 infection-related general practice consultations, hospital admissions including secondary bacterial infections, emergency admissions, out of hours consultations and deaths due to therapies, vaccines and antimicrobials; and (C) adverse events related to therapies, for example, vaccine, antimicrobial administration or other therapies.

## Statistical analysis

Baseline characteristics of all study participants will be described in relation to the study's exposures and outcomes of interest. Mean, median, proportions, ORs and rate ratios (RRs), together with a measure of dispersion will be provided where appropriate to describe differences between the various study groups based on the nature of each variable. The amount of missing data will be described for each variable. Two-tailed hypotheses tests with a 5% significance level will be used for all study's outcomes. All analyses will be carried out using the R statistical programming language.[17–21]

## Primary analyses
### Epidemiology and healthcare burden of COVID-19

We will determine the epidemiological risk factors such as demographic, socioeconomic and clinical population characteristics in relation to laboratory and serology confirmed SARS-CoV-2 infection. The healthcare burden of COVID-19 in terms of morbidity and mortality in relation to to demographic, socioeconomic and clinical population characteristics will also be determined. SARS-CoV-2 infection will be confirmed via laboratory (RT-PCR) and serology testing. Healthcare burden will be measured via general practice consultations, out-of-hours consultations, A&E attendances, hospital admissions including secondary bacterial infections and deaths. Exposure of interest as per our objectives a and b will change over time as the medical literature and surveillance reporting is continuously updated. Currently, particularly factors of interest for Scotland include: age; sex, geographical location, SES, underlying condition or medication and BMI. Analytical techniques including descriptive analysis, univariable and multivariable logistic regression will be used to determine the association between different exposure variables and the likelihood (odds) of the study's primary outcomes (SARS-CoV-2 infection, morbidity, mortality and healthcare burden). The effect of confounders and effect modifiers will be explored through causal frameworks generated for each hypothesis,[33] with clinical input.

## Secondary analyses
### Vaccine uptake

Differences in vaccine uptake will be measured in relation to demographic, socioeconomic and clinical population characteristics. As per primary analyses, exposure of interest will change over time as the medical literature

and surveillance reporting is continuously updated. Key sociodemographic and clinical factors will be analysed including age, sex, SES and underlying condition. Analytical techniques including univariable and multivariable logistic regression will be used to determine the association between different exposure variables and vaccine uptake. The effect of confounders and effect modifiers will be explored through causal frameworks generated for each hypothesis,[33] with clinical input. Key confounding factors will include age, sex, SES and underlying condition. The number of individuals that refuse to be vaccinated and the reasons for declining vaccination will also be investigated, if possible.

### Effectiveness of new or existing prophylactic and therapeutic interventions

We will assess the effectiveness of any new or repurposed therapies, vaccines and antimicrobials against SARS-CoV-2 related morbidity and mortality such as general practice and out-of-hours consultations, hospitalisations including secondary bacterial infections, emergency admissions and deaths. Exposure of interest (therapies, vaccines and antimicrobials) will change over time as the medical literature and surveillance reporting is continuously updated. The proportion of SARS-CoV-2 related clinical outcomes and deaths will be estimated between vaccinated and unvaccinated cases. Vaccine effectiveness (VE) and 95% CIs will be calculated using the formula, VE = (1−risk ratio)*100 for unadjusted and adjusted VE estimates. A time-dependent Cox model or the equivalent Poisson regression models (taking into account the time at risk and the possibility of multiple events (not for death)) will provide the RRs and 95% CIs of VE for prevention of SARS-CoV-2 related clinical outcomes and deaths. Causal frameworks will be generated for each hypothesis,[33] with clinical input. Key confounders for the VE models will include age, sex, SES and underlying condition, with vaccination group representing a time-dependent covariate. In these VE models, propensity variables related to vaccine receipt and effect modifiers (eg, vaccinations, consultations and hospitalisation in the previous season, urban/rural status, smoking status, Charlson Score and pregnancy) will be used to control for the healthy vaccine effect.[17] This is in addition to the demographic variables, which will always be used.

Similar statistical methods will be used to assess the protective effects of therapies and antimicrobials. A binary variable of ever/never exposure to therapies/antimicrobials as an explanatory variable will be included in the VE analyses. The therapy/antimicrobial exposure will be a second time-dependent exposure for consultation, hospitalisation and death rates analysis. We will also consider using a measure of the volume of therapy/antimicrobial exposure (eg, length or dose of prescription) if the data are adequate. Use of therapies/antimicrobials will be included as a covariate in any of our models where primarily assess VE. Alternatively, exposure to the vaccine will be included in any of our models where primarily assess the effect of therapies/antimicrobials, if

appropriate. For example, the effect of therapies/antimicrobials may be assessed from a period before the vaccine becomes available and, in such instances, no adjustment needs to be made.

### Safety of new or existing prophylactic and therapeutic interventions

We will determine any adverse events following the administration of new or repurposed therapies, vaccines and antimicrobials. Specific therapies, vaccines and antimicrobials against SARS-CoV-2 will be determined as the outbreak unfolds and depending on existing medical literature. The risk of adverse events will be estimated using self-controlled study designs. The main assumption in these study designs is that in case of an adverse event related to prophylactic and therapeutic agent exposure, then the occurrence of an adverse event in the period after administration is greater than in periods in the same patients that are temporally not related to prophylactic and therapeutic agent administration.[21] The risk interval (the period at risk for an adverse outcome) and the control interval (the period not at risk for an adverse outcome) will be determined separately for each outcome.[21] Causal frameworks will be generated for each hypothesis,[33] with clinical input. The main advantage of the self-controlled design is the control for all fixed individual-level confounding since any comparisons are carried out for the same individual rather than between exposed and unexposed populations to therapies, vaccines or antimicrobials.[21] Key confounding and effect modifiers will be determined as the outbreak unfolds and depending on existing medical literature.

### Sample size

Our prospective cohort will be constructed from patients registered in all general practices across Scotland with a combined list size of 5.4 million people of all ages. Sample size calculations to assess vaccine and antiviral effectiveness against pandemic influenza have been provided in previous work.[17] Similar sample size calculations are likely to be applicable to the current COVID-19 pandemic; however, sample size calculations (one per key analysis) are dependent on how the COVID-19 outbreak unfolds in Scotland. Thus, our power to answer each objective will be dependent on the frequency of the relevant outcome. Power calculations will be carried out subsequent to the first wave of the pandemic.

### Patient and public involvement (PPI)

We will convene a virtual panel of PPI members who will contribute to the interpretation and dissemination of findings.

### Ethics and dissemination

This study was approved by the National Research Ethics Service Committee, South East Scotland 02. Findings from this study will be presented at international conferences and published in peer-reviewed journals. Metadata produced in this study will also become available to

Health Data Research UK Gateway through BREATHE – The Health Data Research Hub for Respiratory Health. Strengthening the Reporting of Observational Studies in Epidemiology and RECORD (via the COVID-19 extension) will be used to guide transparent reporting.

**Author affiliations**
[1]Wellington School of Health, Faculty of Health, Victoria University of Wellington, Wellington, New Zealand
[2]Usher Institute, The University of Edinburgh, Edinburgh, UK
[3]Department of Mathematics and Statistics, University of Strathclyde, Glasgow, UK
[4]Public Health Scotland, Glasgow, UK
[5]West Of Scotland Specialist Virology Centre, Glasgow, UK
[6]Centre of Academic Primary Care, University of Aberdeen, Aberdeen, UK
[7]The Centre for Health Science, Albasoft Ltd, Inverness, UK

**Acknowledgements** The authors would like to thank and acknowledge Kenny Fraser at Triscribe Ltd and Keith Moffat at Public Health Scotland for their support in this study.

**Funding** The EAVE project was funded by the National Institute for Health Research Health Technology Assessment Programme (project number 13/34/14). EAVE II is funded by the Medical Research Council (MR/R008345/1) and supported by the Scottish Government. We also acknowledge the support of HDR UK. HRS is supported by the Medical Research Council (MR/R008345/1).

**Disclaimer** The views and opinions expressed therein are those of the authors and do not necessarily reflect those of the Health Technology Assessment programme, NIHR, NHS or the Department of Health.

**Conflicts of Interest** CRS reports grants from the UK National Institute for Health Research, Medical Research Council and New Zealand Health Research Council, and The Ministry of Business, Innovation and Employment during the conduct of (and related to) the study. CR reports grants from the UK Medical Research Council, CSO during the conduct of (and related to) the study. CR is a member of the Scottish Government's Chief Medical Officer's COVID-19 Advisory Group. He is also a member of the UK SPI-M committee and the Commission Human Medicines COVID-19 Vaccine Safety Working Group. The views represented in this article do not represent the views of the UK or Scottish Government. JM is Incident Director for COVID-19 at Public Health Scotland and reports no conflicts of interest. LDR serves on a number of Scottish Government Advisory Groups, including COVID-19. MW is a member of the SPI-M advisory committee for the UK Government and the Covid-19 Advisory Group for the Scottish Government. DK is a director of Albasoft Ltd and a health informatician providing technical advice and support to the research community. HRS reports grants from the UK Medical Research Council during the conduct of the study. AS is a member of the Scottish Government's Chief Medical Officer's COVID-19 Advisory Group. The views represented in this article do not represent the views of the Scottish Government. EV, RG, LM, DM and JM report no conflicts of interest.

**Patient consent for publication** Not required.

**Provenance and peer review** Not commissioned; externally peer reviewed.

**ORCID iDs**
Eleftheria Vasileiou http://orcid.org/0000-0001-6850-7578
Helen R Stagg http://orcid.org/0000-0003-4022-3447
Diogo Marques http://orcid.org/0000-0002-2556-0923

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
