## [Reviewer comments · BMJ Open]

ARTICLE DETAILS

TITLE (PROVISIONAL)	Early pandemic evaluation and enhanced surveillance of COVID-19 (EAVE II): protocol for an observational study using linked Scottish national data
AUTHORS	Simpson, Colin; Robertson, Chris; Vasileiou, Eleftheria; McMenamin, Jim; Gunson, Rory; Ritchie, Lewis; Woolhouse, Mark; Morrice, Lynn; Kelly, Dave; Stagg, Helen R.; Marques, Diogo; Murray, Josie; Sheikh, Aziz

VERSION 1 – REVIEW

REVIEWER	Steffanie Strathde UCSD, USA
REVIEW RETURNED	04-May-2020

GENERAL COMMENTS	This is a well written, straightforward surveillance protocol for SARS-CoV-2 which is especially timely. The following are minor issues that are intended to improve what is otherwise an excellent manuscript. 1. Suggest tempering the statement on mortality being lower than SARS-Cov-1 and MERS since current mortality rate estimates for SARS-CoV-2 are highly biased. Instead of 'Current data indicate', use "Current data suggest..."2. Could not this protocol be expanded to include assessment of secondary bacterial infections associated with COVID-19, which are estimated to occur in approx. 10% of hospitalized cases?3. Please clarify whether patients who call NHS 24 for an initial assessment are required to give their CHI number, in which case incidence rates of hospitalization can be determined.4. What proportion of the population in Scotland is expected to be covered by this data linkage? Do people who are not Scottish citizens get captured in these datasets? What other sub-populations might be missed?5. Since smoking status is being captured, is it possible to determine if patients use vaping products or alcohol?6. Comorbidities: Suggest that diabetes be differentiated by type 1 vs. type 2. Suggest that tuberculosis be added.7. Vaccine uptake is being assessed but is it possible to record vaccine refusal and reasons for declining given the growing antivaxx movement?8. Are there any assessments of data quality?
--

REVIEWER	Duorui Shi LinkDoc Technology Inc., Beijing, China
REVIEW RETURNED	16-May-2020

GENERAL COMMENTS	This study has access to a large amount of detailed real world data and may provide relatively reliable clinical evidence for the treatment of COVID-19. I just have 2 minor comments. 1. INTRODUCTION: Line 21. Does "more than a third of the global population infected" exaggerate the fact? This expression is not seen in the reference. And it is not common to replace "global" with "gloabal". 2. METHODS: Exposure definitions and potential confounding factors. Line 8. Please explain how the exposure variables are determined. Ordinary demographic characteristics can hardly cover possible exposure variables. COVID-19 is easy to spread from person to person, so the living environment of the study population, whether it is in contact with the infected person, and daily protective measures should also be considered.
--

VERSION 1 – AUTHOR RESPONSE

Reviewer #1

This is a well written, straightforward surveillance protocol for SARS-CoV-2 which is especially timely. The following are minor issues that are intended to improve what is otherwise an excellent manuscript.

Response: Thank you.

1. Suggest tempering the statement on mortality being lower than SARS-Cov-1 and MERS since current mortality rate estimates for SARS-CoV-2 are highly biased. Instead of 'Current data indicate', use "Current data suggest..."

Response: We have now tempered this statement which reads as follows (see page 4):

"Current data suggest that SARS-CoV-2 has a lower mortality rate, ranged between 0.25% to 3%, than for SARS-CoV (10%) and Middle East Respiratory Syndrome-related coronavirus (MERS-CoV) (37%), respectively.[12-13]"

2. Could not this protocol be expanded to include assessment of secondary bacterial infections associated with COVID-19, which are estimated to occur in approx. 10% of hospitalized cases?

Response: Thank you for this helpful suggestion. We have now expanded our list of outcomes to include assessment of secondary bacterial infections associated with COVID-19 (see pages 7-9):

"The primary outcomes of this study will include: a) laboratory confirmed SARS-CoV-2; b) serum from blood samples taken from biochemistry tests (or rapid antibody tests if available) will be used to determine exposure to SARS-CoV-2 infection by the presence of antibodies; and c) SARS-CoV-2 infection related clinical outcomes including general practice, COVID centres and out-of-hours consultations, hospital admissions including secondary bacterial infections, emergency admissions, out of hours consultations and deaths. Secondary outcomes include: a) vaccine uptake proportions; b) prevention and reduction of SARS-CoV-2 infection-related general practice consultations, hospital admissions including secondary bacterial infections, emergency admissions, out of hours consultations and deaths due to therapies, vaccines and antimicrobials; and c) adverse events related to therapies – e.g. vaccine, antimicrobial administration or other therapies."

“Healthcare burden will be measured via general practice consultations, out-of-hours consultations, A&E attendances, hospital admissions including secondary bacterial infections and deaths.”

“We will assess the effectiveness of any new or repurposed therapies, vaccines and antimicrobials against SARS-CoV-2-related morbidity and mortality such as general practice and out of hours consultations, hospitalisations including secondary bacterial infections, emergency admissions and deaths.”

3. Please clarify whether patients who call NHS 24 for an initial assessment are required to give their CHI number, in which case incidence rates of hospitalization can be determined.

Response: When a patient calls NHS 24 they are not required to give their CHI numbers; they do however give their forename, surname, date of birth and home address. Based on this information NHS 24 callers can instantly identify the patient in their electronic system which already contains the patient’s CHI number. Information that NHS 24 collects from patients is also provided in this link: <https://www.isdscotland.org/Health-Topics/Emergency-Care/Patient-Pathways/>

4. What proportion of the population in Scotland is expected to be covered by this data linkage? Do people who are not Scottish citizens get captured in these datasets? What other sub-populations might be missed?

Response: Our study aims to extract primary care data from all patients registered with a general practice in Scotland. This will lead to over 90% coverage of the Scottish population based on data quality assessment studies that report on completeness of capture of contacts and accuracy of clinical event coding among general practices in Scotland. Non-Scottish citizens can also be captured in these datasets as long as they have a CHI number. An individual obtains a CHI number when they register to their local general practice or after their first healthcare encounter. Non-Scottish residents can also have a CHI number allocated to them, if required. We have thus provided the following statements (page 5):

“Therefore, our study aims to collect data from all residents in Scotland registered with a general practice which translates to over 91% coverage of the Scottish population.[21]”

“A CHI number is also allocated to patients that may have no number when present for treatment as the CHI number is mandatory for all clinical communications. Thus, non-Scottish patients and other temporary residents can also have a CHI number allocated, if required however wherever possible temporary patients will be excluded from this analysis.[22]”

5. Since smoking status is being captured, is it possible to determine if patients use vaping products or alcohol?

Response: We will try to determine these, if possible. The following statement is now provided (page 7):

“The type of smoking products (e.g. vaping products) and alcohol use will also be determined, if possible.”

6. Comorbidities: Suggest that diabetes be differentiated by type 1 vs. type 2. Suggest that tuberculosis be added.

Response: Diabetes is now differentiated into type 1 and 2. Tuberculosis is now also added. Please see the following statement (page 7):

“The following clinical at-risk conditions will be considered: a) chronic respiratory disease (with chronic obstructive pulmonary disease and asthma as subsets); b) chronic heart disease; c) chronic liver disease; d) chronic kidney disease; e) chronic liver disease; f) chronic neurological disease; g) diabetes type 1 and 2; h) conditions or medications causing impaired immune function; i) pregnancy; j) asplenia or dysfunction of spleen; k) obesity (body mass index (BMI) < 20, 20-25, 25-30, 30-39, ≥ 40 kg/m²) l) hypertension (subsets controlled/uncontrolled hypertension); m) tuberculosis and n) multimorbidity.[20]”

7. Vaccine uptake is being assessed but is it possible to record vaccine refusal and reasons for declining given the growing antivaxx movement?

Response: We can provide the number of people that refuse to be vaccinated, but we may be unable to identify the exact reasons that an individual declined the vaccine based solely on administrative healthcare data. We thus have provided the following statement (page 9):

“The number of individuals that refuse to be vaccinated and the reasons for declining vaccination will also be investigated, if possible.”

8. Are there any assessments of data quality?

Response: References on the data quality of primary and secondary care data are now included. We have provided the following statements (page 6):

“Previous observational studies have shown over 91% completeness of capture of contacts and accuracy of clinical event coding (Read codes) among practices in Scotland.[21]”

“Regular validation checks are applied to the SMR database. The latest data quality assessment of these SMR datasets have shown over 90% completeness and accuracy in consistency with previous years.[27]”

Reviewer #2

This study has access to a large amount of detailed real world data and may provide relatively reliable clinical evidence for the treatment of COVID-19. I just have 2 minor comments.

1. INTRODUCTION: Line 21. Does "more than a third of the global population infected" exaggerate the fact? This expression is not seen in the reference. And it is not common to replace "global" with "gloabal".

Response: We have now amended this statement and provided a supportive reference for this. Please see the following statement (page 4):

“In 2009-10, the fourth recorded influenza pandemic due the influenza A (H1N1) subtype emerged in Mexico, resulting in more than 200,000 deaths globally and approximately of 11% to 21% the global population infected.[2, 6]”

2. METHODS: Exposure definitions and potential confounding factors. Line 8. Please explain how the exposure variables are determined. Ordinary demographic characteristics can hardly cover possible exposure variables. COVID-19 is easy to spread from person to person, so the living environment of

the study population, whether it is in contact with the infected person, and daily protective measures should also be considered.

Response: The exposure variables were determined based on a combination of latest COVID-19 evidence and standard exposures that are usually reported in infectious diseases epidemiology. These exposure variables will be further refined or expanded as more evidence arises from the literature. We agree with the Reviewer that given the increased transmission rate of COVID-19 additional exposure factors should be included. We have thus provided the following statement (pages 7-8):

“The effect of population density will also be investigated. Additional exposures such as number of household members for those with a confirmed SARS-CoV-02 infection and daily protective measures will also be investigated given the high transmission rate of COVID-19.”

Editorial comments:

1. Please revise the ‘Strengths and limitations’ section of your manuscript (after the abstract) so that each point consists of a single sentence.

Response: We have now ensured that each point consists of a single point (page 3).

2. Required amendments will be listed here; please include these changes in your revised version:

- Indicate the Corresponding author

Please indicate in the main document file the corresponding author. Kindly amend accordingly.

Response: We have now indicated the corresponding author (see page 1).

3. Complete manuscript information:

- Please complete the “Manuscript information” in ScholarOne submission system (ex: number of tables, figures, supplementary files).

Response: We have now completed the “Manuscript information” in ScholarOne submission system.

4. Figure resolution:

- Please re-upload your figure in 300 dpi and 90mm x 90mm of width. Please see the following link for further details on preparing images for submission:

<https://authors.bmj.com/writing-and-formatting/formatting-your-paper/>

Response: Our figure is now in 300 dpi and 90mm x 90mm of width.

5. Incomplete contributorship statement:

- Please provide a more detailed contributorship statement. It needs to mention all the names/initials of authors along with their specific contribution/participation for the article. *Colin Simpson, Eleftheria Vasileiou, Lewis D. Ritchie, Mark Woolhouse, Lynn Morrice, Dave Kelly, Helen R. Stagg, Diogo Marques, and Josie Murray not mentioned in contributorship statement

Response: We have now provided a detailed contributorship statement with all authors’ initials included. We have also corrected Colin Simpson’s initials which are now Colin R Simpson (see page 11).

Additional changes: We have also added Keith Moffat in our acknowledgements as we accidentally omitted to include him in our previous submission. (see page 11).

We trust that these revisions are to your satisfaction; please do not however hesitate to contact us if you need any further clarification or revisions.

VERSION 2 – REVIEW

REVIEWER	Steffanie Strathdee UC San Diego, USA
REVIEW RETURNED	29-May-2020

GENERAL COMMENTS	This is a very thorough response. My only remaining suggestion is that in the case of secondary bacterial infections that are sequelae of COVID19, the authors attempt to capture which bacterial infections are MDR. There is a growing consensus that the COVID pandemic will worsen the superbug crisis and these data will be helpful in the response to both.
--

REVIEWER	Duorui Shi LinkDoc Technology Inc., Beijing, China
REVIEW RETURNED	30-May-2020

GENERAL COMMENTS	The authors have responded to my comments. I have no further comments.
--

VERSION 2 – AUTHOR RESPONSE

Reviewer #1

This is a well written, straightforward surveillance protocol for SARS-CoV-2 which is especially timely. The following are minor issues that are intended to improve what is otherwise an excellent manuscript.

Response: Thank you.

1. Suggest tempering the statement on mortality being lower than SARS-Cov-1 and MERS since current mortality rate estimates for SARS-CoV-2 are highly biased. Instead of 'Current data indicate', use "Current data suggest..."

Response: We have now tempered this statement which reads as follows (see page 4):

"Current data suggest that SARS-CoV-2 has a lower mortality rate, ranged between 0.25% to 3%, than for SARS-CoV (10%) and Middle East Respiratory Syndrome-related coronavirus (MERS-CoV) (37%), respectively.[12-13]"

2. Could not this protocol be expanded to include assessment of secondary bacterial infections associated with COVID-19, which are estimated to occur in approx. 10% of hospitalized cases?

Response: Thank you for this helpful suggestion. We have now expanded our list of outcomes to include assessment of secondary bacterial infections associated with COVID-19 (see pages 7-9):

“The primary outcomes of this study will include: a) laboratory confirmed SARS-CoV-2; b) serum from blood samples taken from biochemistry tests (or rapid antibody tests if available) will be used to determine exposure to SARS-CoV-2 infection by the presence of antibodies; and c) SARS-CoV-2 infection related clinical outcomes including general practice, COVID centres and out-of-hours consultations, hospital admissions including secondary bacterial infections, emergency admissions, out of hours consultations and deaths. Secondary outcomes include: a) vaccine uptake proportions; b) prevention and reduction of SARS-CoV-2 infection-related general practice consultations, hospital admissions including secondary bacterial infections, emergency admissions, out of hours consultations and deaths due to therapies, vaccines and antimicrobials; and c) adverse events related to therapies – e.g. vaccine, antimicrobial administration or other therapies.”

“Healthcare burden will be measured via general practice consultations, out-of-hours consultations, A&E attendances, hospital admissions including secondary bacterial infections and deaths.”

“We will assess the effectiveness of any new or repurposed therapies, vaccines and antimicrobials against SARS-CoV-2-related morbidity and mortality such as general practice and out of hours consultations, hospitalisations including secondary bacterial infections, emergency admissions and deaths.”

3. Please clarify whether patients who call NHS 24 for an initial assessment are required to give their CHI number, in which case incidence rates of hospitalization can be determined.

Response: When a patient calls NHS 24 they are not required to give their CHI numbers; they do however give their forename, surname, date of birth and home address. Based on this information NHS 24 callers can instantly identify the patient in their electronic system which already contains the patient’s CHI number. Information that NHS 24 collects from patients is also provided in this link: <https://www.isdscotland.org/Health-Topics/Emergency-Care/Patient-Pathways/>

4. What proportion of the population in Scotland is expected to be covered by this data linkage? Do people who are not Scottish citizens get captured in these datasets? What other sub-populations might be missed?

Response: Our study aims to extract primary care data from all patients registered with a general practice in Scotland. This will lead to over 90% coverage of the Scottish population based on data quality assessment studies that report on completeness of capture of contacts and accuracy of clinical event coding among general practices in Scotland. Non-Scottish citizens can also be captured in these datasets as long as they have a CHI number. An individual obtains a CHI number when they register to their local general practice or after their first healthcare encounter. Non-Scottish residents can also have a CHI number allocated to them, if required. We have thus provided the following statements (page 5):

“Therefore, our study aims to collect data from all residents in Scotland registered with a general practice which translates to over 91% coverage of the Scottish population.[21]”

“A CHI number is also allocated to patients that may have no number when present for treatment as the CHI number is mandatory for all clinical communications. Thus, non-Scottish patients and other temporary residents can also have a CHI number allocated, if required however wherever possible temporary patients will be excluded from this analysis.[22]”

5. Since smoking status is being captured, is it possible to determine if patients use vaping products or alcohol?

Response: We will try to determine these, if possible. The following statement is now provided (page 7):

“The type of smoking products (e.g. vaping products) and alcohol use will also be determined, if possible.”

6. Comorbidities: Suggest that diabetes be differentiated by type 1 vs. type 2. Suggest that tuberculosis be added.

Response: Diabetes is now differentiated into type 1 and 2. Tuberculosis is now also added. Please see the following statement (page 7):

“The following clinical at-risk conditions will be considered: a) chronic respiratory disease (with chronic obstructive pulmonary disease and asthma as subsets); b) chronic heart disease; c) chronic liver disease; d) chronic kidney disease; e) chronic liver disease; f) chronic neurological disease; g) diabetes type 1 and 2; h) conditions or medications causing impaired immune function; i) pregnancy; j) asplenia or dysfunction of spleen; k) obesity (body mass index (BMI) < 20, 20-25, 25-30, 30-39, ≥ 40 kg/m²) l) hypertension (subsets controlled/uncontrolled hypertension); m) tuberculosis and n) multimorbidity.[20]”

7. Vaccine uptake is being assessed but is it possible to record vaccine refusal and reasons for declining given the growing antivaxx movement?

Response: We can provide the number of people that refuse to be vaccinated, but we may be unable to identify the exact reasons that an individual declined the vaccine based solely on administrative healthcare data. We thus have provided the following statement (page 9):

“The number of individuals that refuse to be vaccinated and the reasons for declining vaccination will also be investigated, if possible.”

8. Are there any assessments of data quality?

Response: References on the data quality of primary and secondary care data are now included. We have provided the following statements (page 6):

“Previous observational studies have shown over 91% completeness of capture of contacts and accuracy of clinical event coding (Read codes) among practices in Scotland.[21]”

“Regular validation checks are applied to the SMR database. The latest data quality assessment of these SMR datasets have shown over 90% completeness and accuracy in consistency with previous years.[27]”

9. This is a very thorough response. My only remaining suggestion is that in the case of secondary bacterial infections that are sequelae of COVID19, the authors attempt to capture which bacterial infections are MDR. There is a growing consensus that the COVID pandemic will worsen the superbug crisis and these data will be helpful in the response to both.

Response: We agree with the reviewer that identifying the Multidrug Resistant (MDR) bacteria associated with the secondary bacterial infections for COVID19 is pertinent. We thus have provided the following statement (page 8):

“The primary outcomes of this study will include: a) laboratory confirmed SARS-CoV-2; b) serum from blood samples taken from biochemistry tests (or rapid antibody tests if available) will be used to determine exposure to SARS-CoV-2 infection by the presence of antibodies; and c) SARS-CoV-2 infection related clinical outcomes including general practice, COVID centres and out-of-hours consultations, hospital admissions including secondary bacterial infections and Multidrug Resistant

(MDR) bacteria associated with these infections, emergency admissions, out of hours consultations and deaths.”

Reviewer #2

This study has access to a large amount of detailed real world data and may provide relatively reliable clinical evidence for the treatment of COVID-19. I just have 2 minor comments.

1. INTRODUCTION: Line 21. Does "more than a third of the gloabal population infected" exaggerate the fact? This expression is not seen in the reference. And it is not common to replace "global" with "gloabal".

Response: We have now amended this statement and provided a supportive reference for this. Please see the following statement (page 4):

“In 2009-10, the fourth recorded influenza pandemic due the influenza A (H1N1) subtype emerged in Mexico, resulting in more than 200,000 deaths globally and approximately of 11% to 21% the global population infected.[2, 6]”

2. METHODS: Exposure definitions and potential confounding factors. Line 8. Please explain how the exposure variables are determined. Ordinary demographic characteristics can hardly cover possible exposure variables. COVID-19 is easy to spread from person to person, so the living environment of the study population, whether it is in contact with the infected person, and daily protective measures should also be considered.

Response: The exposure variables were determined based on a combination of latest COVID-19 evidence and standard exposures that are usually reported in infectious diseases epidemiology. These exposure variables will be further refined or expanded as more evidence arises from the literature. We agree with the Reviewer that given the increased transmission rate of COVID-19 additional exposure factors should be included. We have thus provided the following statement (pages 7-8):

“The effect of population density will also be investigated. Additional exposures such as number of household members for those with a confirmed SARS-CoV-02 infection and daily protective measures will also be investigated given the high transmission rate of COVID-19.”

Editorial comments:

1. Please revise the ‘Strengths and limitations’ section of your manuscript (after the abstract) so that each point consists of a single sentence.

Response: We have now ensured that each point consists of a single point (page 3).

2. Required amendments will be listed here; please include these changes in your revised version:

- Indicate the Corresponding author

Please indicate in the main document file the corresponding author. Kindly amend accordingly.

Response: We have now indicated the corresponding author (see page 1).

3. Complete manuscript information:

- Please complete the "Manuscript information" in ScholarOne submission system (ex: number of tables, figures, supplementary files).

Response: We have now completed the "Manuscript information" in ScholarOne submission system.

4. Figure resolution:

- Please re-upload your figure in 300 dpi and 90mm x 90mm of width. Please see the following link for further details on preparing images for submission:

<https://authors.bmj.com/writing-and-formatting/formatting-your-paper/>

Response: Our figure is now in 300 dpi and 90mm x 90mm of width.

5. Incomplete contributorship statement:

- Please provide a more detailed contributorship statement. It needs to mention all the names/initials of authors along with their specific contribution/participation for the article. *Colin Simpson, Eleftheria Vasileiou, Lewis D. Ritchie, Mark Woolhouse, Lynn Morrice, Dave Kelly, Helen R. Stagg, Diogo Marques, and Josie Murray not mentioned in contributorship statement

Response: We have now provided a detailed contributorship statement with all authors' initials included. We have also corrected Colin Simpson's initials which are now Colin R Simpson (see page 11).

6. We note that in the Patient and Public Involvement statement it is stated that you will pursue the involvement of patients or the public in the research study. Please be more specific in your plans to

involve patients or the public. If patients or the public have not been involved in the study design/preparation of the study protocol, then please clarify how they will be involved in the conduct of the study. If there are no plans to involve patients or the public in the design or conduct of this study then please state this. Please see our Instructions for Authors for further details:

https://bmjopen.bmj.com/pages/authors/#reporting_patient_and_public_involvement_in_research

Please note that the Patient and Public Involvement statement should be placed at the end of the methods section.

Response: We have now described the involvement of patients and the public in our study. We have thus provided the following statement at the end of the methods section (pages 10):

“Patient and public involvement

We will convene a virtual panel of PPI members who will contribute to the interpretation and dissemination of findings.”

7. Please rephrase the first sentence of the ethics and dissemination section in your main text to specifically state that the study was approved by the National Research Ethics Service Committee, South East Scotland 02.

Response: We have now rephrased this sentence and provided the following statement (page 10):

“This study was approved by the National Research Ethics Service Committee, South East Scotland 02.”

Additional changes: We have also added Keith Moffat in our acknowledgements as we accidentally omitted to include him in our previous submission. (see page 11).

We trust that these revisions are to your satisfaction; please do not however hesitate to contact us if you need any further clarification or revisions.